# Therapy-Induced Senescent/Polyploid Cancer Cells Undergo Atypical Divisions Associated with Altered Expression of Meiosis, Spermatogenesis and EMT Genes

**DOI:** 10.3390/ijms23158288

**Published:** 2022-07-27

**Authors:** Joanna Czarnecka-Herok, Malgorzata Alicja Sliwinska, Marcin Herok, Alicja Targonska, Anna Strzeszewska-Potyrala, Agnieszka Bojko, Artur Wolny, Grazyna Mosieniak, Ewa Sikora

**Affiliations:** 1Laboratory of Molecular Bases of Ageing, Nencki Institute of Experimental Biology, Polish Academy of Sciences, 02-093 Warsaw, Poland; joanna.czarnecka-herok@lyon.unicancer.fr (J.C.-H.); marcin.herok@gmail.com (M.H.); a.targonska@nencki.edu.pl (A.T.); a.starzeszewska@nencki.edu.pl (A.S.-P.); a.bojko@nencki.edu.pl (A.B.); g.mosieniak@nencki.edu.pl (G.M.); 2Laboratory of Imaging Tissue Structure and Function, Nencki Institute of Experimental Biology, Polish Academy of Sciences, 02-093 Warsaw, Poland; m.sliwinska@nencki.edu.pl (M.A.S.); a.wolny@nencki.edu.pl (A.W.)

**Keywords:** SBEM scans, cancer, cell senescence, chemotherapy, polyploidization, soma-to-germline transition, senescence escape

## Abstract

Upon anticancer treatment, cancer cells can undergo cellular senescence, i.e., the temporal arrest of cell division, accompanied by polyploidization and subsequent amitotic divisions, giving rise to mitotically dividing progeny. In this study, we sought to further characterize the cells undergoing senescence/polyploidization and their propensity for atypical divisions. We used p53-wild type MCF-7 cells treated with irinotecan (IRI), which we have previously shown undergo senescence/polyploidization. The propensity of cells to divide was measured by a BrdU incorporation assay, Ki67 protein level (cell cycle marker) and a time-lapse technique. Advanced electron microscopy-based cell visualization and bioinformatics for gene transcription analysis were also used. We found that after IRI-treatment of MCF-7 cells, the DNA replication and Ki67 level decreased temporally. Eventually, polyploid cells divided by budding. With the use of transmission electron microscopy, we showed the presence of mononuclear small cells inside senescent/polyploid ones. A comparison of the transcriptome of senescent cells at day three with day eight (when cells just start to escape senescence) revealed an altered expression of gene sets related to meiotic cell cycles, spermatogenesis and epithelial–mesenchymal transition. Although chemotherapy (DNA damage)-induced senescence is indispensable for temporary proliferation arrest of cancer cells, this response can be followed by their polyploidization and reprogramming, leading to more fit offspring.

## 1. Introduction

Cellular senescence is the cell fate that affects both normal and cancer cells exposed to harmful agents, including those damaging DNA. Senescent cells do not divide but stay alive and change their metabolism and phenotype [1]. The most remarkable feature of senescent cells is the so-called senescence-associated secretory phenotype (SASP) [2]. The cytokines and other molecules secreted by senescent cells profoundly affect the cellular and tissue microenvironment and exert mostly, but not exclusively, deleterious effects, including pro-cancerogenic ones [3]. Anticancer therapy, applied both in vitro and in vivo, has been shown to induce cellular senescence [4]. This type of cellular senescence became known as therapy-induced senescence (TIS) [5]. Initially, TIS was considered beneficial because of the lower drug doses needed to restrict tumor growth by inducing senescence. Later, however, partially due to SASP, TIS was categorized as a mechanism of cancer resistance and/or dormancy [6]. Meanwhile, it emerged that anticancer treatment can lead to polyploidization of cancer cells [7]. Interestingly, several groups have independently observed that polyploid giant cancer cells (PGCCs), rather than using the traditional mode of mitotic division, can generate progeny cells via amitotic mechanisms, including budding, splitting or bursting [8,9,10,11,12,13,14]. The coexistence of these two phenomena, namely cancer cell polyploidization on one hand, and cancer cell senescence on the other hand, suggests that they must be closely related. Indeed, there is growing evidence that polyploid cancer cells that arise from treatment with anticancer drugs have the characteristics of senescent cells [15]. PGCCs, which have their own life cycle and are now thought to play an important role in the origin, immortality, invasion, metastasis, and resistance of tumor cells to radiotherapy and chemotherapy, show features of senescence [6,10,16,17,18,19,20,21,22]. Senescent cells are usually identified by a set of common markers. These include: increased activity of lysosomal SA-β-galactosidase, DNA damage foci and key DNA damage response (DDR) proteins (ATM, p53), increased level of cell cycle inhibitors (p16 and p21) and changed morphology (enlarged size and granularity) [23].

In this study, we attempted to better characterize the nature of senescent/polyploid cancer cell divisions and other features differing them from parental cells. To this end, we used the same approach as in our previous study [16]. We treated MCF-7 (p53-positive), MDA-MB-231(triple negative) breast cancer cells and HCT116 (p53-positive) colon cancer cells with commonly used chemotherapeutics: irinotecan (IRI), doxorubicin (DOXO) and methotrexate (MTX). However, most experiments were performed on IRI-treated MCF-7 cells.

We have shown for the first time, based on transmission electron microscopy analysis, the origin of descendants of senescent cells, which regain proliferation potential. Those newborn cells appeared to arise inside the PGCCs. The process of escaping from senescence was associated with senescent cell reprogramming, leading to changes in the expression of the genes involved in meiosis and spermatogenesis regulation, as documented by the transcriptome analysis. We may speculate that those changes are necessary for the division of polyploid/senescent cells.

## 2. Results

### 2.1. Resumption of Proliferation of Senescent/Polyploid MCF-7 Cells

We have previously documented that treatment of different types of cancer cell lines with various anticancer drugs led to cell senescence [16] and polyploidization [15]. This is also true for MCF-7 cells treated with irinotecan (IRI). In the current study, we used the same protocol, namely one-day drug administration followed by several days of culture in fresh medium (Appendix A). Of note, such a protocol, at least in the case of doxorubicin, is clinically relevant [24]. Previously, we have shown the increase in several markers of senescence in MCF-7 cells treated with 5 µM IRI: SA-β-gal activity, cell granularity, cell size, the p53/p21 signaling pathway, the IL-8 and VEGF signaling pathway and IL-8 and VEGF secretion [16]. Now, we show that the IRI-treated MCF-7 cell population underwent temporal inhibition of proliferation (Figure 1a), which was confirmed by a decrease in DNA synthesis (BrdU incorporation assay) and a decrease in the number of Ki67-positive cells on the third day after treatment (D3) (Figure 1b,c). At D8, DNA synthesis and cell number increased again. The polyploid cells accounted for more than one-third of all cells at D3 and D8 (Figure 1a). Because Ki67 is a cell cycle marker which is absent in cells that are in G0, we conclude that the majority of cells at D3 do not divide and are unlikely to resume divisions. However, some giant polyploid cells are Ki67-positive (Figure 1d) and we suspect that these cells give rise to a progeny. Interestingly, on D3, practically only the giant polyploid cells were BrdU-positive, while there were only less than 2% of cells with small nuclei, which were able to synthetize DNA (Figure 1b). It suggests that the increase in proliferating cells on D8 cannot be related to the cells that did not undergo senescence (Figure 1b).

This is a fairly common phenomenon, similar to that seen in cells of other cancer cell lines, in which proliferation is temporarily arrested after various treatments (Appendix A). In our previous work, we demonstrated the formation of numerous subnuclei after doxorubicin treatment of HCT116 colon cancer cells [12]. We observed the same phenomenon in MCF-7 cells treated with IRI. To test the integrity of the nuclear envelope of the resulting subnucleus, we stained cells for A/C lamin. Multinucleated cells were shown to retain nuclear membrane integrity in both the large and small subnucleus, as we have previously observed for HCT 116 cells treated with doxorubicin [12]. This result also indicates that these cells are not dying (Figure 1d).

### 2.2. Atypical Morphology of MCF-7 Senescent/Polyploid Cells

The serial block-face scanning electron microscopy (SBEM) method, which allows for three-dimensional reconstruction and in-depth observation of the nucleus structure, was applied. Scans obtained with this method were used to prepare a three-dimensional reconstruction of senescent/polyploid cell nucleus (Figure 1e, right panel), which highlights its complex and irregular structure with numerous cavities. Interestingly, the nucleus of the untreated cell also had an irregular structure; however, the presence of small subnuclei was not observed (Figure 1e, left panel).

### 2.3. Atypical Divisions of Senescent/Polyploid MCF-7 Cells

Using scanning electron microscopy, we showed that doxo-treated HCT 116 cells [11] and IRI-treated MCF-7 cells are flattened and giant (Figure 2a). Some much smaller spherical cells sit on them, which can indicate that they are the offspring of giant cells. Previously, we demonstrated atypical divisions of doxorubicin-treated HCT 116 cells [12]. We have now documented the budding of IRI-treated MCF-7 cells (Figure 2b).

Subsequently, transmission electron microscopy was used to visualize the process of atypical cell divisions (Figure 3).

As the divisions of the polyploid cells are quite rare events, several hundred cells were imaged at various time points to record the appearance of the first small proliferating cells. Here, we present images which may suggest that inside the giant mother cell there are small cells. Actin filaments can be seen surrounding the subnucleus and mitochondria at the periphery of the polyploid cell (Figure 3a). Of particular note is a separate small cell surrounded by the polyploid cell, for which the cell membrane is in contact with the membrane of the polyploid cell (Figure 3a, blue arrow). The second phenomenon is the formation of a subnucleus inside the giant cell, with mitochondria surrounded by a membrane (Figure 3b). The membrane of the small cell was visible in different sections of the same area (Figure 3b, z axis). In Figure 3c a subnucleus and mitochondria at the periphery of the cell can be seen detaching from the cell along with a small amount of cytoplasm.

### 2.4. Transcriptome Analysis

We performed RNA sequencing to reveal differences between the transcriptome of untreated cells and senescent/polyploid cells at D3 and D8, when senescent/polyploid cells start to divide [25] (Figure 4).

Gene set enrichment analysis (GSEA) revealed positive enrichment of gene sets related to the cellular response to lipids in D3 cells compared with control (Figure 4a), together with genes related to stemness characteristics such as epithelial to mesenchymal transition, ABC transporters and drugs metabolism (Figure 4b). Moreover, senescent/polyploid cancer cells are characterized by lipid droplet accumulation (Figure 4a, black arrow), and glycogen accumulation (Figure 4a, blue arrow) can be seen. A comparison of the transcriptome between control–D3–D8 revealed the altered expression of genes related to the meiotic cell cycle, spermatogenesis and EMT in D3 and D8 cells (Figure 4c). We discovered a few genes expressed only in D8 cells that were related to the meiotic cell cycle: MYBL1, C19orf57, DUSP13, REC8 and BCL2L11. The D8 cell population comes back to some extent to the proliferative state, as shown by the mitotic cell cycle and E2F targets (Appendix A); however, it goes together with the altered expression of genes related to meiosis, spermatogenesis and EMT (Figure 4c).

Overall, our analyses revealed many differences between senescent/polyploid cells on D3 and D8, which will be discussed in the following section.

## 3. Discussion

In this study, for the first time, we documented that escape from senescence/polyploidy of cancer cells is connected with the presence of small descendants inside the giant cells and the transcriptome gradual adaptation related to meiosis and spermatogenesis. Polyploid cell divisions have already been described in the literature. However, the relationship between polyploidization and senescence is still an unresolved issue—in particular, the presentation of unequivocal evidence for the division of a polyploid senescent cancer cell. The phenomenon of division of polyploid cells occurs quite rarely, according to some authors, with a frequency of 1 in 10^6^ [25]. The low frequency of this process makes it very difficult to capture, and time-lapse techniques and fluorescence microscopy cannot distinguish without any doubt whether a mononuclear cell is under, above or inside the polyploid/senescent cell.

Whole-genome duplication (polyploidization) in cancer cells is a very well recognized phenomenon [26]. A polyploid genome has been found in 37% of solid tumors [27]. In fact, polyploid giant cancer cells (PGCCs) are the driving force behind not only tumorigenesis but also metastasis [28]. They facilitate rapid tumor evolution and the acquisition of therapy resistance in multiple incurable cancers [27,29]. Even one senescent PGCC can induce a metastatic tumor when grafted into animals [14]. DNA damaging agents used in cancer therapy are common inducers of cancer cell polyploidy [19,30,31]. Similar to cancer cell polyploidization, cancer cell senescence has also been documented in many cell lines upon anticancer therapy [32]. In addition, both radiotherapy [33] and chemotherapy [6] can quite easily trigger the senescence response in cancer cells in vivo [34,35]. Eventually, it was found that the same agents can induce both senescence and polyploidization in different cancer cell types, so these two processes must be related [15].

In our previous studies, we used mainly doxorubicin as an inducer of cancer cell senescence/polyploidization [11,12,15,20]. Recently, in our laboratory, senescence has also been induced in various cancer cells (such as A549, SH-SY-5Y, HCT116, MDA-MB-231 and MCF-7) through treatment with other anticancer drugs, such as methotrexate, 5-fluorouracil, oxaliplatin, or paclitaxel [16]. In the current study, we used breast cancer cells (MCF-7) undergoing senescence after treatment with irinotecan [16] and have documented that, already, at D3 after pulse treatment, there was almost 40% polyploid cells (>4c DNA). Slightly more polyploid cells were observed at D8. The cell number at D8, in comparison to D3, increased substantially. At D3, there were also substantially less BrdU-positive and Ki67-positive cells than at D8, indicating that several days after treatment the cells intensified DNA synthesis and divisions and effectively escaped from senescence.

It was originally shown that Ki67 is only expressed in cycling cells [36] and is a common marker of cancer cells [37]. Senescent cells are presumed to be Ki67-negative [38]. However, it has been documented that Ki67 expression during the cell cycle is not simply binary but is relatively high in G2 and becomes gradually attenuated after mitosis when cells are entering G1 [38]. Thus, Ki67-negative cells should not be definitely considered as noncycling, as they can also be in the G1 phase of the cell cycle (with hardly detectable levels of Ki67). It cannot be excluded that, when senescent cells are in the cell cycle, they may replicate their DNA but skip or slip mitosis, resulting in polyploidy, and then return to the proliferative state through depolyploidization. Thus, we have postulated that senescent cancer cells do not exit the cell cycle [1]. The low number of IRI-treated MCF-7 Ki67-positive (mitotically active) cells at D3 may indicate that, at that time, the majority of cells are in G1 after slipping mitosis (Ki67-negative). Indeed, Salmina et al. recently documented that senescent/polyploid cells are slipping mitosis (endocycling) and subsequently generating a progeny [20]. Interestingly, at D3, we observed rare multinucleated cells in which either the main nucleus or many subnuclei were Ki67-positive. We suppose that these rare Ki67-positive cells are dividing in an atypical way. Previously, we have shown that Ki67-positive giant cells are also SA-β-gal-positive, thus contradicting the notion that senescent cells are Ki67-negative and out of the cell cycle [39]. However, at D8, the cell number increased, as did the number of cells that were BrdU- and Ki67-positive. This indicates that their high proliferation capacity derives mainly from cells which have escaped from senescence. It is reasonable to postulate that senescent/polyploid cells cannot replicate DNA and increase their size indefinitely; thus, they either have to die or divide.

Many studies have documented that senescent cancer cells can divide atypically and give mitotically dividing progeny [15,19]. Although therapeutic strategies promoting cell senescence are still under debate, it seems that the concept that cellular senescence represents a state of permanent growth arrest and a barrier to cancer development [40] is not a valid paradigm in the case of cancer cells. Accumulating evidence suggests that cells have the capacity to escape TIS, which supports the conclusion that senescence allows the cells to enter a temporary state of dormancy that eventually facilitates disease recurrence, often in a more aggressive state [41].

Our model of senescence/polyploidy revealed positive enrichment of ABC transporters, drug metabolism and epithelial–mesenchymal transition gene sets at D3 in comparison to control cells, showing that there is a specific type of reprograming in cells undergoing senescence/polyploidy. There are studies, including ours, which show that some of the characteristics of stemness are associated with induced senescence/polyploidy in cancer cells [10,21,22,42,43]. Moreover, Milanovic et al. [44] documented that cancer cells induced to senescence were temporally arrested in the cell cycle and subsequently, after reprogramming, acquired features of stemness and re-entered the cell cycle. Post-senescent cells became more aggressive than their presenescent counterparts. The characteristics of stemness were demonstrated as increased aldehyde dehydrogenase (ALDH) and ATP-binding cassette (ABC) transporter activities. However, reversible polyploidy formation was not considered in that study [44]. Furthermore, Weihua et al. showed that polyploid cells from untreated cultures were SA-β-gal-positive and were also able to grow as spheroids and give rise to tumor and metastases [14]. Erenpreisa and Cragg suggested that, paradoxically, it may be necessary for a cancer cell to undergo senescence in order to rejuvenate after genotoxic treatments. Resistance to anticancer treatment is thus provided through a programmed life-cycle-like process, which closely links senescence, polyploidy and stemness [45]. However, there is a study in which purified and cultured SA-β-gal-negative PGCCs derived from human ovarian cancer cells started to divide asymmetrically and acquired cancer stem cell markers. A single PGCC formed cancer spheroids in vitro and generated tumors in immunodeficient mice. These PGCC-derived tumors gained a mesenchymal phenotype with increased expression of the cancer stem cell markers CD44 and CD133 [29].

Atypical divisions of polyploid giant cells were first postulated by Erenpreisa’s [8] and Rajaraman’s [13] groups. According to Sundaram et al. [13] the giant polyploid cells give rise to tiny Raju cells by neosis, which resembles yeast budding. In turn, Erenpreisa claimed that divisions of giant cells resembled the process of meiosis [46]. The results presented in this paper show that these two processes (budding and reductional divisions) are not necessarily mutually exclusive. First, with the use of SBEM combined with three-dimensional reconstruction, we show 3D structures of many subnuclei located peripheral to the larger nucleus of senescent/polyploid cell. These subnuclei are large enough that they could be a separate nucleus of the emerging small cells. Division of senescent polyploid cells occurs quite rarely, so the imaging of several hundred cells was carried out at the time when the first small proliferating cells began to appear (D8).

For the first time, by using sophisticated visualization methods, we evidenced that inside the giant mother cells there are small entities surrounded by actin fibers and a membrane. We identified three phenomena that show possible ways for progeny cells to form from senescent polyploid cancer cells. However, it should be noted that they do not provide unequivocal evidence for the emergence of small living cells through the occurrence of these processes. Asymmetric cytokinesis of multinucleated cells without preceding mitosis was postulated by a Sundaram et al.; however, this work did not provide evidence for senescence of the observed cells, and the surviving descendants were characterized by the lack of the presence of the p53 protein [13]. More unequivocal results on the occurrence of senescent cell divisions (the process of polyploidization was not analyzed) was provided by Saleh et al., who observed senescence after etoposide treatment of nonsmall cell lung H460 cancer cells, in which a regime of sorting based on C12FDG activity and cell size was used. The sorted cells returned to dividing in an in vitro model, as well as when implanted into immunodeficient mice [47]. Moreover, with the use of time-lapse experiments, we showed that small cells are released from giant mother cells. The giant cells are multinucleated and some of the subnuclei may possess a proper set of chromosomes, which allows them to survive. Our observations support the idea of polyploid cells escaping from senescence through amitotic divisions.

Liu laboratory documented that PGCCs continued self-renewal via endoreplication and further divisions by nuclear budding or fragmentation; the small daughter nuclei then acquired cytoplasm, split off from the giant mother cells and acquired competency in mitosis [48]. On the other hand, we showed here that senescent/polyploid cells starting to escape senescence (D8) have enriched expression not only of mitotic, but also of some meiotic cell cycle genes. Notably, we discovered a few genes expressed only in cells at D8 which are related to the meiotic cell cycle: MYBL1, C19orf57, DUSP13, REC8 and BCL2L11. Previously, it was documented that lymphoma p53-mutated cell lines after irradiation underwent depolyploidization leading to production of mitotically dividing cells [8], and it was accompanied by the expression of MOS, REC8, DMC1, STAG3, SYCP3 and SYCP1 [49,50]. Recently, Salmina et al. [20] showed that senescent cancer cells in repeated mitotic slippage cycles activated the meiotic genes MOS, REC8 and SPO11 and displayed holocentric chromosomes which are characteristic for so-called inverted meiosis [51]. These giant cells acquired an amoeboid phenotype and finally depolyploidized by budding, producing a mitotically dividing progeny [20].

In this study, gene set enrichment analysis revealed both meiotic and mitotic gene alterations (D8). Thus, we suppose that, at that time, in the entire cell population there are still senescent/polyploid cells undergoing reductional divisions, as well as their mitotically dividing progeny. Interestingly, at D3 and D8, when compared to control cells, we also observed an enrichment in the genes involved in spermatogenesis. These data support the concept that genetic alterations in cancer can result in the activation of normally silent germline expression programs, and that these programs might confer some of the central characteristics of malignancy. Germline genes include genes for so-called cancer/testis (CT) antigens, which are normally expressed by gametes and trophoblasts but are aberrantly expressed in a range of human cancers. Some CT antigens are encoded by genes located on the X chromosome (CT-X antigens) and are usually highly expressed in the spermatogonia–mitotically proliferating germ cells. The CT-X genes in cancer cells control gene expression and directly influence cell proliferation and the sensitivity of cancer cell lines to cytotoxic assault [52]. Because the donor of the breast MCF7 cancer cells was female, it would be expected that the CT-X gene expression would be increased after irinotecan treatment in this cell line. However, more analyses are needed to prove this. Especially, single-cell RNA-seq analysis should be performed to assign transcriptome changes to descendants or mother cells, as the use of bulk RNA-seq cannot answer which population is enriched with particular genes. Nonetheless, we can speculate that enrichment in the expression of meiotic genes and the genes involved in spermatogenesis supports the model of soma-to-germline transition as a hallmark of human aggressive cancers [53,54]. Moreover, it appears that any insult to cancer cells that leads to reversible senescence/polyploidy is a way to accelerate cancer evolution.

We speculate that stress generated by chemotherapeutics or irradiation can induce either senescence or polyploidization, or both of these processes. Depending on the cell type and stress, increased ploidy can precede the senescence onset or, quite the reverse, the senescence phenotype can be followed by polyploidization. However, it seems that polyploidy rather than senescence is indispensable for cancer relapse. Xu et al. reported that histone acetylase inhibitor-induced cell senescence was followed by polyploidy formation, leading eventually to the generation of small progeny [55]. Recently, we have shown that doxorubicin treatment of MDA-MB-231 cells induced senescence followed by a ploidy increase and the appearance of progeny [17]. Moreover, we have documented previously that an antioxidant, Trolox, reduced the number of polyploid cells without affecting senescence of doxorubicin-treated HCT 116 cells and that this protected the cells against escape from senescence as the colony formation assay revealed [11]. Thus, it can be concluded that polyploidization is indispensable for the division of cancer cells, which is combined with an escape from senescence, because polyploid cells have the characteristics of cell senescence. Thus, overall, it seems rather impossible to separate the role of these two processes, namely polyploidization and senescence, in producing the progeny.

Our research revealed another hallmark common to cellular senescence and polyploidy at D3—the accumulation of lipid droplets (LDs), which has recently been shown to be a characteristic feature of normal [56] and cancer [17] senescent cells. In healthy cells, free fatty acids are directly used for energy production, so lipid synthesis is usually inactive. Cancer cells, which are constantly faced with various stressors, such as hypoxia of the tumor microenvironment, nutrient fluctuations or anticancer treatment, accumulate LDs. An increase in LDs can affect major cancer features, including cell growth, proliferation, metabolism, migration, inflammation, immunity, aggressiveness and chemotherapy resistance [57]. Interestingly, polyploid cancer cells have greater accumulation of LDs than nonpolyploid cancer cells [57,58]. In this study, we visualized the accumulation of LDs in senescent/polyploid cancer cells and showed the increased expression of genes involved in responses to lipids (D3). Moreover, glycogen accumulation has been observed in replicative senescence of human primary fibroblasts and in the tissues of the liver of old rats [59]. Here, we show that MCF-7 cells treated with IRI accumulate glycogen as well. Both excessive lipogenesis and glycogenesis are associated with impaired metabolism of the cell undergoing senescence. Glycogen granules may be at least partially responsible for the increase in granularity of MCF-7 after IRI treatment. The role of glycogen as a store of energy in the senescence process is not elucidated and requires further research [59].

Altogether, an interesting feature of PGCCs emerges from the data presented in this study. PGCCs are significantly enlarged, flattened, giant and multinucleated cells. They are characterized by a temporary stop in the cell cycle followed by atypical divisions, gradual alterations of the expression of meiotic genes and the genes involved in spermatogenesis and stemness. Moreover, ultrastructure analysis revealed the possible origin of progeny cells, which formed inside some PGCCs. In addition, some of their features overlap with those attributed to senescent (normal and cancer) cells: damage to DNA, increased granularity and activity of SA-β-gal, accumulation of LDs, accumulation of glycogen and reprogramming. However, it seems that these features of senescent cells are merely a side effect and do not alter the immanent property of PGCCs, namely their ability to undergo aberrant cell divisions to generate a more aggressive progeny. On the other hand, we can speculate that senescence of cancer cells is indispensable for temporary cell cycle arrest, which gives time for reprogramming to produce a more fit offspring. Thus, it can be said that polyploid cancer cell senescence and dormancy represent the same physiological state [6].

## 4. Materials and Methods

### 4.1. Reagents

Doxorubicin (D1515), irinotecan hydrochloride (I1406) and methotrexate (M9929) were purchased from Sigma-Aldrich (Saint Louis, MO, USA).

### 4.2. Culture of Cancer Cells

Human colon cancer HCT116 (CCL-247) cell line was kindly provided by Dr. Bert Vogelstein (Johns Hopkins University, Baltimore, MD, USA). Breast cancer MCF-7 cell line (HTB-22) was purchased from the American Type Culture Collection (ATCC). Cells were grown under standard conditions (37 °C, 5% CO_2_) in McCoy’s (HCT116), DMEM low-glucose (MCF-7) medium supplemented with 10% fetal bovine serum, 100 units/mL of penicillin, 100 µg/mL of streptomycin and 25 µg/mL amphotericin B. To induce senescence, cancer cells were seeded at a density of 10,000/cm^2^ 24 h before treatment with a chemotherapeutic. Next, cancer cells were incubated with 2.5 µM (HTC116), 5 µM (MCF-7) irinotecan or 100 nM doxorubicin or 1.25 µM (HTC116), 2.5 µM (MCF-7) methotrexate for 24 h. Cells were analyzed in terms of senescence markers three days after drug removal.

### 4.3. Immunocytochemistry

Staining was performed as described in Piechota et al. [58]. Briefly, cells were washed with PBS and fixed in 4% paraformaldehyde for 10 min. Cell membranes were permeabilized by 10 min incubation in 0,5% Triton X-100 in PBS, then cells were blocked in 2% BSA, 1.5% goat serum and 0.1% Triton X-100 in PBS for 10 min. Afterwards, cells were incubated on slides with Ki67 antibody (Abcam, Cambridge, UK) 1:500, lamin a/c antibody (Cell signaling, Leiden, Netherlands) in blocking buffer (2% BSA, 1.5% goat serum and 0.1% Triton X-100 in PBS). Next, secondary Alexa 488-conjugated IgG antibody (1:500) or 555-conjugated IgG antibody (1:500) was used (Life Technology, Warsaw, Poland). To visualize the nuclei, DNA was stained using Hoechst dye (2 μg/mL in PBS) (Life Technology, Warsaw, Poland). The cells were analyzed with the Zeiss LSM800 confocal microscope, with excitation at 488 nm and emission at 495–550 nm for Alexa Fluor 488 and excitation at 568 nm, and emission at 580–650 nm for Alexa Fluor 568. Excitation at 405 nm and emission at 414–471 nm was used for detection of Hoechst.

### 4.4. Bromodeoxyuridine Incorporation Assay

Cells, seeded on glass coverslips in 12-well plates, were treated with 10uM BrdU for the last 18 h of culture. Next, they were washed with PBS and fixed in ice-cold 70% ethanol. Cells were washed with 0.5% TritonX-100 (Sigma-Aldrich, Poznan, Poland) in PBS, incubated in 2N HCl for 30 min, washed twice with PBS, incubated for 1 min in 0.1 M borax solution (Sigma-Aldrich, Poznan, Poland), washed twice in PBS again and incubated with primary antibody (Beckton Dickinson, Franklin Lakes, NJ, USA) 1:120 in 1% BSA 0.5% Tween-20 solution in PBS for 1 h. Then, cells were washed twice with 0.5% Tween-20 in PBS and incubated with secondary antibody conjugated to a fluorochrome (Thermo Fisher Scientific, Warsaw, Poland). Cells were washed with 0.5% Tween-20 in PBS, stained with 1M DAPI for 15 min, washed again and mounted on a microscope slide with FluoroMount medium (Thermo Fisher Scientific, Warsaw, Poland). Specimens were examined under Nikon Eclipse fluorescent microscope. Images were analyzed using the ImageJ program. The number of BrdU-positive cells was calculated relative to the number of all cells (based on DAPI staining).

### 4.5. DNA Content Estimation by Flow Cytometry

For DNA analysis, the cells were fixed in 70% ethanol and stained with PI solution (3.8 mM sodium citrate, 500 µg/mL RNAse A and 50 µg/mL PI in PBS). All agents were purchased from Sigma-Aldrich (Poznan, Poland). DNA content was assessed using flow cytometry (FACSCalibur with CellQuest Software—Becton Dickinson, Franklin Lakes, NJ, USA). Ten thousand events were collected per sample.

### 4.6. EM Sample Preparation

Cells growing on glass coverslips were fixed using a mixture of 2% paraformaldehyde (Sigma-Aldrich, P6148), 1% glutaraldehyde (EMS, EM grade) and 2mM CaCL2 in 0.2 M Hepes pH 7.3 and were prepared for electron microscopy according to a published protocol [60] with minor changes [61]. Briefly, cells were postfixed with 1% aqueous solution of osmium tetroxide (Agar Scientific, AGR1023) and 1.5% potassium ferrocyanide (Sigma-Aldrich, St. Louis, MO, USA, P3289) in phosphate buffer for 30 min on ice, immersed in 1% aqueous thiocarbohydrazide (Sigma-Aldrich, St. Louis, MO, USA) for 20 min, postfixed with 2% aqueous solution of osmium tetroxide for 20 min (all at room temperature) and incubated in 1% aqueous solution of uranyl acetate at 4 °C overnight. The next day, samples were exposed to 0.66% lead nitrate in aspartic acid for 30 min at 60 °C, dehydrated with increasing concentrations of ethanol, infiltrated with Durcupan resin (Sigma-Aldrich, St. Louis, MO, USA, #44610), embedded using BEEM capsules according to a published protocol [61] and cured at 70 °C for 72 h.

For transmission electron microscopy, ultrathin sections (65 nm thick) were cut with an ultramicrotome (ultracut R, Leica, Microsystems, Vienna, Austria) and collected on formvar-coated 100 mesh copper grids (Agar Scientific, AGS138-1).

For serial block-face scanning electron microscopy, a piece of resin with cells was mounted to aluminum pins (Gatan system pins, Micro to Nano, Netherlands, 10-006003-50) with cyanoacrylate glue and trimmed. The face of the block was polished and grounded with a silver paint (Ted Pella, Redding, CA, USA).

For scanning electron microscopy, samples were fixed using a mixture of 2% paraformaldehyde (Sigma-Aldrich, P6148), 1% glutaraldehyde (EMS, EM grade) and 2mM CaCL2 in 0.2 M Hepes pH 7.3. They were then dehydrated with graded dilutions of ethanol, dried using critical point drier (Polaron, United Kingdom) and sputter coated with thin layer of gold (JEOL Co., Japan, Tokyo).

### 4.7. Transmission Electron Microscopy Imaging

Specimen grids were examined with a JEM 1400 (JEOL Co., Tokyo, Japan, 2008) transmission electron microscope, equipped with an 11 megapixel TEM camera MORADA G2 (EMSIS GmbH, Münster, Germany).

### 4.8. Serial Face-Block Scanning Electron Microscopy Imaging

Samples were imaged using Zeiss Sigma VP scanning electron microscope (Zeiss, Oberkochen, Germany) equipped with 3View2 GATAN chamber using a backscatter electron detector. Imaging settings: magnification 15,000×, variable pressure 13 Pa, EHT 3 kV, aperture 20 μm, pixel dwell time 5 μs, pixel size 6.3–7.7 nm and section thickness 100 nm.

### 4.9. View Scan Processing and Image Analysis

Stacks of images were aligned using the ImageJ software (ImageJ-> Plugins-> Registration-> StackReg) and saved as .tiff image sequence. After alignment, scans were imported to the Reconstruct software Version 1.1.0.0, 2007, creator John C. Fiala, Ph.D., Boston University, Boston, MA, U.S.A., available at http://synapses.clm.utexas.edu/tools/reconstruct/reconstruct.stm (accessed from 1 July to 1 December 2019). For 3D reconstructions, a selected brick was chosen from each sample and Ľ of its volume (5 μm × 5 μm × 1.5 μm) was subjected to 3D reconstructions in Imaris software.

### 4.10. RNA Isolation and Sequencing

RNA was isolated using RNeasy^®^ Plus Mini Kit (Qiagen, Venlo, The Netherlands). RNA integrity was checked using the RNA Nano 6000 Assay Kit of the Bioanalyzer 2100 system (Agilent Technologies, Santa Clara, CA, USA) and concentration was measured with Qubit^®^ RNA Assay Kit in Qubit^®^ 2.0 Fluorometer (Life Technologies, Carlsbad, CA, USA). Stranded mRNA-Seq libraries were prepared from 300 ng of total RNA using the Illumina TruSeq RNA Sample Preparation v2 Kit (Illumina, San Diego, CA, USA), implemented on the liquid handling robot Beckman FXP2. Obtained libraries that passed the QC step, which was assessed on the Agilent Bioanalyzer system, were pooled in equimolar amounts. A 1.8 pM solution of each pool of libraries was loaded on the Illumina sequencer NextSeq 500 High output and sequenced unidirectionally, generating ~450 million reads per run, each 85 bases long. All samples were aligned to the GRCh38 genome assembly using the STAR RNA-seq aligner. The EMBL Genomics Core Facility (Heidelberg, Germany) performed library preparation, RNA sequencing and alignment.

### 4.11. Bioinformatics Analysis

Aligned gene counts were analyzed using the edgeR package (Bioconductor release version 3.10) in R software environment (version 3.6.2). The cpm function from the edgeR was used to normalize the different sequencing depths for each sample. Trimmed mean of M values (TMM) normalization was performed to eliminate composition biases between libraries [62]. The voom function from the limma package (version 3.44.3) was used to transform the read counts into logCPMs (counts-per-million) while taking into account the mean–variance relationship in the data [63]. Then, voom-transformed data were fit into a linear model to test for differentially expressed genes. Gene set enrichment analysis (GSEA) [64] was performed using fgsea algorithm [65]. Input for the analysis was a list of all the genes which passed quality control, sorted from most significantly upregulated to most significantly downregulated genes and the input annotation of term-to-gene mapping was downloaded in version v7.0 from https://www.gsea-msigdb.org (accessed on 8 December 2019), provided by the Broad Institute and UC San Diego (H collection of hallmark gene sets, C2 collection of curated gene sets—CP KEGG and REACTOME databases, as well as selected C2 subcollections CGP: Chemical and Genetic Perturbations, as indicated in the figure descriptions). Normalized enrichment scores (NES) with FDR-adjusted *p*-values < 0.05 were considered statistically significant. Plots were prepared using plotEnrichment function from fgsea package. Pathview package [66] was used to render gene expression data on relevant KEGG database pathway graphs by converting it to pseudocolors on a scale from the most repressed (blue, −2 in arbitrary units) to most upregulated (yellow, +2 in arbitrary units). Other visualizations were performed with ggplot2 and pheatmap packages.

### 4.12. Live Imaging

Holographic microscope, HoloMonitor4 (LabSoft, Warsaw, Poland) was used as a life imaging technique. Film acquisition took 3–5 days; time between each frame: 20 min.

### 4.13. Statistical Analysis

Statistical analysis was performed with the use of the STATISTICA 13 program, employing ANOVA, followed by Tukey’s honestly significant difference (HSD) test. A value of *p* < 0.05 was considered statistically significant (*p* < 0.05 *, *p* < 0.01 **, *p* < 0.001 ***). All graphs show the mean results from at least three independent experiments. Error bars represent SEM.

## Figures and Tables

**Figure 1 ijms-23-08288-f001:**
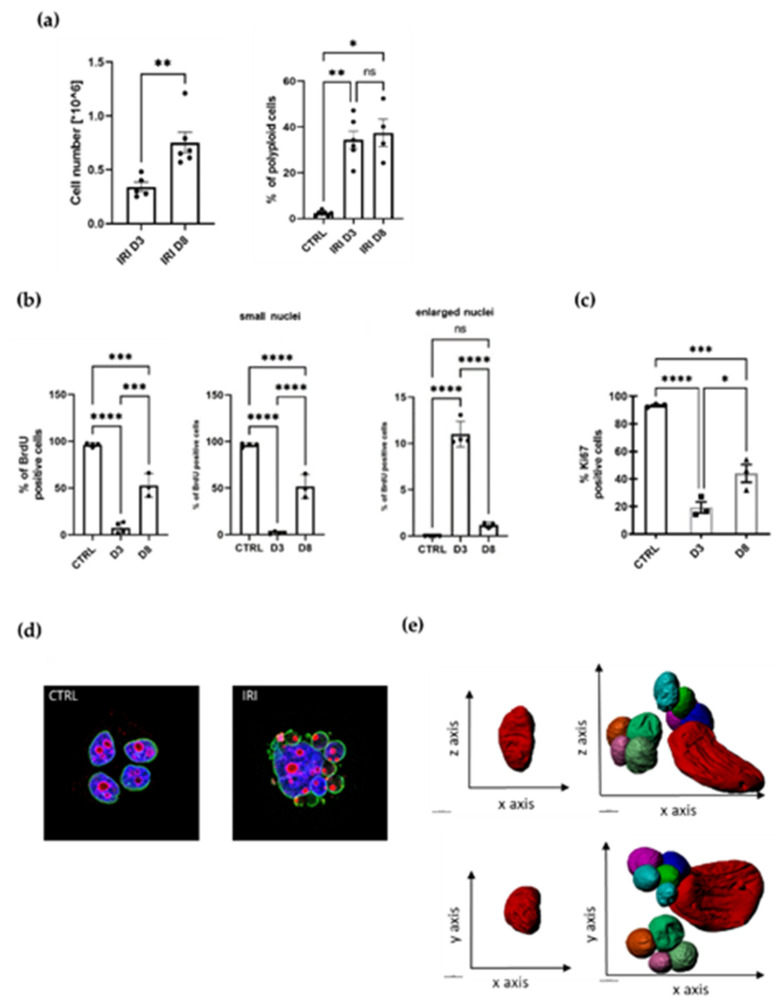
Therapy-induced senescence is associated with increased polyploidization and followed by resumption of proliferation. (**a**) MCF-7 breast cancer cells counted at day three (D3) and day eight (D8) after 24 h treatment with IRI (left panel); quantification of polyploid cells assessed with flow cytometry at day three (D3) and day eight (D8), >4c DNA cells were considered as polyploid (right panel); mean of at least three independent experiments ± SEM; statistical significance: * 0.01 < *p* < 0.05, ** 0.001 < *p* < 0.01. (**b**) Quantification of BrdU-positive MCF-7 cells; mean of at least three independent experiments ± SEM; statistical significance: * 0.01 < *p* < 0.05, ** 0.001 < *p* < 0.01, *** *p* < 0.001, **** *p* < 0.0001 (**c**) Quantification of Ki67-stained MCF-7 cells at day three (D3) and day eight (D8) mean of at least three independent experiments ± SEM; statistical significance: * 0.01 < *p* < 0.05, *** *p* < 0.001, **** *p* < 0.0001. (**d**) Representative images of control and treated cells stained with lamin a/c (green) and Ki67 (red), nuclei stained with Hoechst 33342 (blue). (**e**) 3D reconstruction of the serial block-face scanning electron microscopy (SBEM) sections. Representative reconstruction of control (on the left) and IRI-treated MCF-7 cells at D8 (on the right); scale bar = 50 µM.

**Figure 2 ijms-23-08288-f002:**
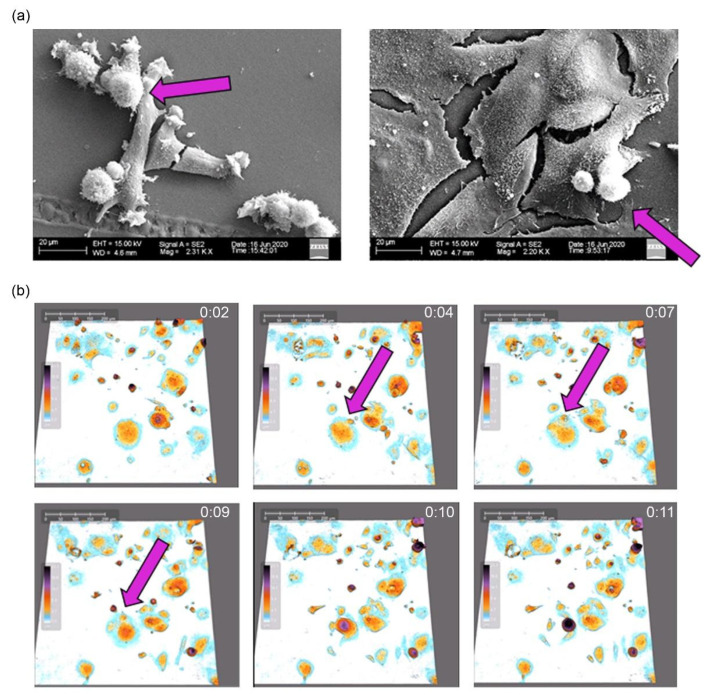
Atypical divisions of senescent/polyploid cancer cells. (**a**) Representative pictures of control MCF-7 breast cancer cells (on the left) and ones after treatment with IRI. Arrow indicates dividing cells. Pictures were taken on D8 with the use of scanning electron microscope. (**b**) Time lapse of asymmetric division of senescent MCF-7 breast cancer cells after IRI treatment was documented with holographic microscope HoloMonitor4. Time between frames: 20 min. Senescent cell with small budding offspring cells (pink arrow).

**Figure 3 ijms-23-08288-f003:**
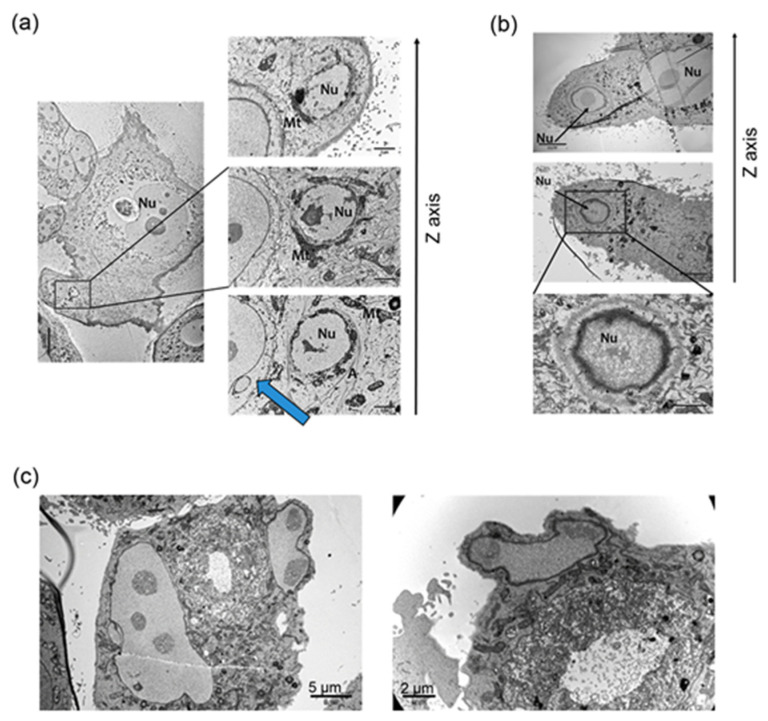
Formation of small offspring cells during escape from TIS. (**a**) Subnucleus surrounded by actin filaments and mitochondria. Blue arrow shows a separate small cancer cell, surrounded by a senescent polyploid one. Representative pictures of senescent and polyploid MCF-7 cells after 24 h of IRI treatment; picture taken on D8 after treatment. (**b**) Formation of a subnucleus inside the senescent cell with mitochondria surrounded by a cell membrane. Representative pictures of senescent and polyploid MCF-7 cells after 24 h of IRI treatment and eight additional days. (**c**) Protrusion of a subnucleus inside the senescent cell with mitochondria. Representative pictures of senescent and polyploid HCT116 cells after 24 h of IRI treatment; picture taken on D8 after treatment. Nu—nuclei, Mt—mitochondria, A—actin. Pictures taken with the use of transmission electron microscopy.

**Figure 4 ijms-23-08288-f004:**
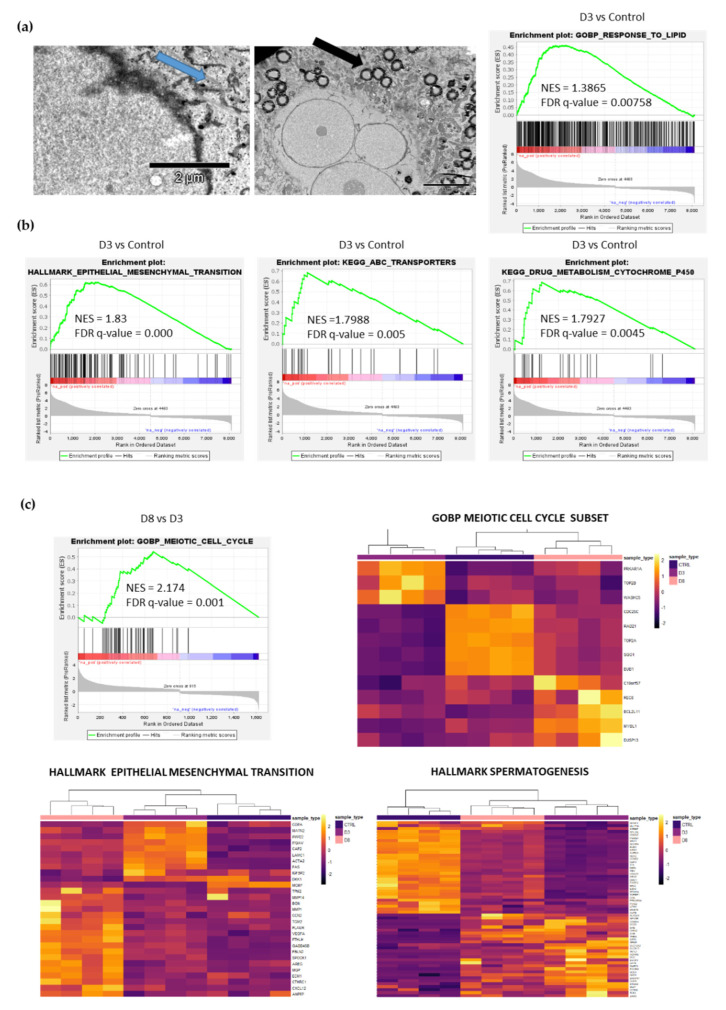
Transcriptome changes during escape from TIS. (**a**) Representative images of glycogen (left panel, blue arrow) and lipid droplets (right panel, black arrow) of IRI-treated senescence MCF-7 cells. Pictures were taken with the use of transmission electron microscope. GSEA analysis of the MCF-7 cells transcriptome on D3 compared to control after IRI. Positive enrichment in the indicated GOBP (gene ontology biological processes) pathway is shown. (**b**) GSEA analysis of the MCF-7 cells transcriptome on D3 compared to control after IRI treatment. Positive enrichment in indicated KEGG and HALLMARK pathways is shown. (**c**) GSEA analysis of the MCF-7 cells transcriptome on D8 compared to D3 after IRI treatment. Positive enrichment in the indicated GO pathway is shown. Heatmaps representing GOBP Meiotic Cell Cycle, HALLMARK EPITHELIAL MESENCHYMAL TRANSITION and HALLMARK SPERMATOGENESIS were prepared on differentially expressed genes from the respective gene set.

## Data Availability

Not applicable.

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
