# Peer review of "Therapy-Induced Senescent/Polyploid Cancer Cells Undergo Atypical Divisions Associated with Altered Expression of Meiosis, Spermatogenesis and EMT Genes"

_ijms, 2022, doi:10.3390/ijms23158288_

Round 1
Reviewer 1 Report
This is a logical follow-up work by this group in which the authors employ advanced single-cell visualization methods to determine the fate of breast cancer cells after treatment with the chemotherapeutic drug irinotecan (IRI). They show that chemotherapy-induced senescence (TIS), which is widely assumed to reflect favorable clinical outcome, is accompanied by polyploidy/multinucleation and emergence of tumor repopulating progeny. The work is extremely important, the experiments are well designed, and the results support the conclusions. Studies such as this (rather than those that use short-term, high content and misleading preclinical assays such as 96-well plate cytotoxicity/viability) is a step forward in the metaphor of the “war on cancer” which isn’t won yet, as highlighted in a recent Nature Editorial (Nature, Vol 601, 20 January 2022, page 297).
I have some suggestions that the authors might want to consider:
It is known that only one polyploid/multinucleated giant cancer cell can form a spheroid in tissue culture (Oncogene 33, 116–128, 201, 2014) and induce metastatic tumor when grafted into animals (Cancer 117, 4092–4099, 2011). Reminding the reader of these properties of PGCCs will further emphasize the clinical relevance of the observations reported in the current submission.
The authors use drug treatment protocol (24 h drug exposure followed by chase in drug-free medium) that is known to be consistent with how drugs are administered to patients. For example, a recent review on DOX (Cancer Chemotherapy and Pharmacology, 89:285–311, 2022) has determined that 24 exposure to 100 nM DOX is clinically relevant. (On the other hand, continuous drug exposures which is widely used in most cytotoxicity/apoptosis studies are clinically irrelevant; see, e.g., Oncotarget 2017, 8, 8854–8866, 2017). Perhaps the authors want to emphasize this important point that they use clinically relevant chemotherapeutic drug treatment.
It will be informative to some readers if the authors would clarify the p53 status of the MCF7 and HCT116 cell lines used in this paper, and perhaps that of the cell lines used in their previous work.
Line 16. Consider saying “We used the p53 wild-type MCF7 breast carcinoma cells…” also inset “to” after “shown”
Line 24. The word “speculate” is probably too weak. I would suggest using something like “Although chemotherapy-induced senescence is indispensable for temporary proliferation arrest of cancer cells, this response can be followed by their polyploidization and reprogramming leading to a more fit offspring.”
Probably there is no need to partition the Abstract into sections; i.e., consider deleting “(1) background…(2) Methods…etc”
Line 184. When referring to different cell lines on line 184 and other places in the text, instead of saying “different cancer cells” (which could mean different cancer cells within an individual tumor), consider saying different cancer cell types or different cancer cell lines.
Is there a way of incorporating the supplementary material (or at least only panel a) in the main text?
The Methods describes three chemotherapeutic drugs where only one (IRI0 is mentioned in the abstract and other places. Is this because the other two are used in the supplementary material? Please clarify.
Author Response
Please, see the attachment

Reviewer 2 Report
The manuscript from Czarnecka-Herok and colleagues describes some long-term outcomes (after 3 and 8 days) of MCF7 cells exposed to IRI as a model of therapy induced senescence. Authors show that under their conditions, treated cells firstly acquire features of senescence (cell cycle arrest, enlargement, polyploidy) at day 3 (D3) before to restart proliferation at D8. They also point out an atypical division of PGCC by budding, and report an increased expression of genes associated with mitosis and meiosis in cells at D8 compared to cells at D3. Although authors are experts in the field and the presented data could be interesting in the assessment of what is really associated with senescence in cancer cells, I cannot clearly find any novelty regarding state of the art. Indeed, all the phenotypes described here have already been reported in previous works, as nicely reviewed in 2021 by authors (first reference of this manuscript).
Major points
The last sentence of the abstract is confusing as DNA damage induces an early and transient cell cycle arrest eventually precede senescence, apoptosis or cell cycle restart.
Figure 1 and S1 b: IRI exposure in these experimental conditions does not seem to completely stop cell proliferation, as evidenced by the presence of BrdU and KI67 positive cells (fig 1 b and c). Therefore, the increase in cell number between D3 and D8 (fig 1a) could result from a minor part of cells that did not halt their cell cycle and/or did not enter senescence, thus being selected contrary to senescent cells. This point is crucial for the interpretation of the following data. In contrast, MCF7 cells exposed to DOXO or MTX do not restart their proliferation (fig S1 b), suggesting that in this context, cells did not escape from senescence-associated cell cycle arrest. How IRI concentration has been selected? At least a colony formation assay with a dose-response must be included to show the proportion of viable cells. Moreover, what is the proportion of BrdU positive polyploid cells?
Figure 2b: what is the time legend? hours? More important: what is the fate of small and giant cells after budding? In the last image, it seems that the giant cell is dying. I understand that this represents a rare event, but this result may greatly consolidate the current findings.
Figure 3: how can the authors exclude that small cells inside GPCC do not result from entosis?
Figure 4: GSEA analyzis should be done between D8 and control to confirm the increase of mitotic/meiotic pathways.
Minor points
There are lot of typographical errors (mainly spacebars).
Author Response
Please, see the attachment.

Reviewer 3 Report
The submitted manuscript provides some interesting propositions related to recovery from polyploidy/senescence. However, it is this reviewer's opinion that the study requires significant additional studies and/or clarifications.
1. I did not see any senescence characterization data. The authors appear to just assume that the polyploid cells they are tracking are also senescent without confirming experimentally. The methods appear to indicate that these experiments were performed (section 4.2.) but I do not see the corresponding data?
2. The methods also indicate that cell cycle and dna index estimation was performed, but again, I do not see this data in main or supp figures?
3. The authors should explain how they see polyploid cells dividing or budding into mononuclear daughter cells, but see an equal percentage of polyploid cells in the population at D8 (which actually would indicate a greater number of polyploid cells since there are more cells overall in the population).
4. This reviewer does not see the significance in the transcriptome comparisons. Besides the EMT signature, most of these gene signatures are going to be related to growth. The authors have demonstrated that the D8 population consists of more actively proliferating cells than the D3 population. So would it not be logical that the differences seen in the D8 population are going to be related to growth? Rather than a mechanism, it appears to be more a validation of the growth recovery. Are the D8 cells significantly different than the parental cells?
5. I find the train of thought in the discussion hard to follow, particularly in regards to the discussion about whether senescence or polyploidy is dispensable.
Author Response
Please, see the attachment.

Round 2
Reviewer 2 Report
In this revised version, authors clarified most of the issues. They emphasized in their response the novelty of their work, but this is still not clear in the manuscript.
One originality of this study is to show for the first time mononuclear cells inside PGCCs by TEM. However, this is not really highlighted. Notably, the discussion section presents a well documented state of the art of senescence escape and cell fate, which is in the scope of the study but not directly related to the novelties presented here.
I still have some problems with transcriptional analyzes, which are over-interpreted to me in the discussion section. Of note, cells at D8 are composed by different cell populations (polyploid cells, small nuclei with or without BrdU), which are, according to the author response, "mainly senescent cells with some proportion of mononuclear proliferating cells". it is therefore difficult to assign transcriptional modification to one specific cell population. However, in the first sentence of the discussion, it is stated that "In this study for the first time we documented that escape from senescence/ polyploidyof cancer cells is connected with the presence of small descendants inside the giant cells and increased transcription of genes involved in meiosis and spermatogenesis", and latter in the discussion that "PGCCs are significantly enlarged, flattened, giant and multinucleated cells. They are characterized by a temporary stop in the cell cycle followed by atypical divisions, increased expression of meiotic genes and genes involved in spermatogenesis and stemness". Concerning meiosis, 38% of genes are upregulated in D8, 38% in CTRL and 23% in D3. In other words, 61% of meiotic genes are not upregulated in D8 in PGCC after senescence escape. Concerning spermatogenesis, more than an half of genes are downregulated in D3 and D8 compared to CTRL, the other part being indeed upregulated, but with not a so marked difference between D3 and D8 (Fig 4C). Considering this, the data do not support that transcriptional changes observed in D8 concern mononuclear dividing cells derived from PCGG after atypical division, and not senescent cells that represent the main population. Giant senescent cells can indeed gradually adapt their transcription profile from D3 to D8.
To conclude, the discussion should be less general on senescence escape and more focused on the present data, especially those representing the originality of the study, and must avoid over-interpretation of the transcriptional alterations observed at D8.
Author Response
In this revision of our manuscript, we have made significant changes in line withthe reviewer's comments. We apologize that not all changes are visible as we have
already accepted some of them for the sake of clarity. We hope this version
of the discussion will be acceptable to the reviewer. Thanks again for your valuable
comments.
Reviewer 3 Report
Acceptable with current changes. Thanks.
Author Response
Thank you.